# Healthcare practitioner experiences and willingness to prescribe pre-exposure prophylaxis in the US

Ashley A. Leech[1,2� *], Cindy L. Christiansen[3☺], Benjamin P. Linas[4,5☺], Donna M. Jacobsen[6☺], Isabel Morin[1☺], Mari-Lynn Drainoni[2,5,7,8☺]

**1** Department of Health Policy, Vanderbilt University School of Medicine, Nashville, TN, United States of America, **2** Department of Health Law, Policy and Management, Boston University School of Public Health, Boston, MA, United States of America, **3** Boston University School of Dental Medicine, Boston, MA, United States of America, **4** Center for Infectious Diseases, Boston Medical Center, Boston, MA, United States of America, **5** Department of Medicine, Section of Infectious Diseases, Boston University School of Medicine, Boston, MA, United States of America, **6** International Antiviral Society-USA, San Francisco, CA, United States of America, **7** Evans Center for Implementation and Improvement Sciences, Department of Medicine, Boston University School of Medicine, Boston, MA, United States of America, **8** Department of Veterans Affairs, Center for Healthcare Organization and Implementation Research, Bedford, MA, United States of America

☺ These authors contributed equally to this work.
* ashley.leech@vanderbilt.edu

**Data Availability Statement:** Anonymized Raw data is available as a supplementary file.

## Abstract

### Background and objectives

Less than 10 percent of the more than one million people vulnerable to HIV are using pre-exposure prophylaxis (PrEP). Practitioners are critical to ensuring the delivery of PrEP across care settings. In this study, we target a group of prescribers focused on providing HIV care and seeking up-to-date information about HIV. We assessed their experiences prescribing PrEP, whether these experiences differed by clinical specialty, and examined associations between willingness to prescribe PrEP as a "best first step" and different hypothetical prescribing scenarios.

### Setting and methods

Between March and May 2015, we circulated a paper survey to 954 participants ((652 of whom met our inclusion criteria of being independent prescribers and 519 of those (80%) responded to the survey)) at continuing medical education advanced-level HIV courses in five locations across the US on practitioner practices and preferences of PrEP. We employed multivariable logistic regression analysis for binary and collapsed ordinal outcomes.

### Results

Among this highly motivated group of practitioners, only 54% reported ever prescribing PrEP. Internal medicine practitioners were 1.6 times more likely than infectious disease practitioners to have prescribed PrEP (95% CI: 0.99–2.60, p = .0524) and age, years of

**Funding:** Benjamin P. Linas and Mari-Lynn Drainoni received funding from the Providence/ Boston Center for AIDS Research (P30AI042853). No other authors received funding for this work. The funders had no role in study design, data collection and analysis, decision to publish, or preparation of the manuscript.

**Competing interests:** The authors have declared that no competing interests exist.

training, and sex were significantly associated with prescribing experience. Based on clinical vignettes describing different hypothetical prescribing scenarios, practitioners who viewed PrEP as the first clinical step for persons who inject drugs (PWID) were twice as likely to have also considered PrEP as the first clinical option for safer conception, and vice-a-versa (95% CI: 1.4–3.2, p < .001). Practitioners considering PrEP as the first preventive option for MSM were nearly six times as likely to also consider PrEP as the first clinical step for PWID, and vice-a-versa (95% CI: 2.28–13.56, p = .0002).

## Conclusions

Our findings indicate that even among a subset of HIV-focused practitioners, PrEP prescribing is not routine. This group of practitioners could be an optimal group to engage individuals that could most benefit from PrEP.

## Introduction

HIV prevention in the US has received renewed political attention with heightened goals to eradicate the virus by 2030 [1]. The latest US Preventive Services Task Force recommendation on pre-exposure prophylaxis (PrEP) underscores its important role in this effort [2]. The once-daily medications are recommended for men who have sex with men (MSM), persons who inject drugs (PWID), and heterosexually active men and women who are behaviorally vulnerable to HIV [2, 3]. However, only 8 percent of the estimated 1.2 million people vulnerable to HIV are using PrEP [2, 4, 5].

Structural barriers, such as practitioner prescribing practices and comfort with PrEP, could be important influences on PrEP implementation across clinical settings. The "purview paradox," i.e. the notion that neither HIV practitioners nor primary care physicians consider PrEP to fall within their clinical purview, may explain the slow adoption of PrEP among practitioners [6–13]. Given PrEP is to be used by individuals who do not have HIV, primary care and family physicians likely have a central role in facilitating access to PrEP among individuals particularly vulnerable to the virus.

Treating different priority groups also adds another layer of complexity to practitioner PrEP prescribing practices. For example, studies have found varying levels of confidence and willingness among practitioners to prescribe PrEP across patient groups, with practitioners expressing greater inclination to prescribe to MSM than to behaviorally vulnerable heterosexuals or PWID [14–16]. Moreover, routine risk assessments occur infrequently in primary care, which can influence practitioner prescribing of PrEP if patients are not assessed for HIV risk [11, 17].

In this study, we target a group of prescribers focused on providing HIV care and seeking up-to-date information about HIV (as evidenced by their attendance of an advanced HIV continuing medical education course). We assessed their experiences prescribing PrEP, whether these experiences differed by clinical specialty, and examined associations between willingness to prescribe PrEP as a "best first step" and different hypothetical prescribing scenarios.

We hypothesized that there would be differences in experience prescribing PrEP by practitioner specialty (infectious disease, internal medicine, family medicine) and between practitioner opinions on offering PrEP as a "best first step" and different hypothetical risk scenarios

((MSM, sero-different heterosexuals seeking conception (comprised of one person with HIV and one without HIV), and PWID)).

## Methods

### Study design

In collaboration with the International Antiviral Society-USA (IAS-USA), between March and May 2015, we circulated a voluntary paper survey to 954 participants at continuing medical education (CME) advanced-level HIV courses in five locations across the US: San Francisco, CA, Los Angeles, CA, New York, NY, Washington, DC, and Chicago, IL. We surveyed participants before the course session on PrEP. We did not provide incentives for survey completion. We included in our sample only practitioners who could independently prescribe, including physicians ((Doctors of Medicine [MDs]) or Doctors of Osteopathic Medicine [DOs]), nurse practitioners (NPs), and physician assistants (PAs). The Boston University Medical Center Institutional Review Board (IRB) determined this study to be HIPPA exempt in accordance with 45 CFR 46.101. As per IRB stipulations, we included a consent statement at the top of the survey form.

We used $\alpha = 0.05$ for all statistical tests and set $\beta$ at 0.20 (or a power of 0.80) when assessing sample size, which indicated adequate power at a total sample size of 450 to detect small effects (0.14–0.15) between prescriber categories (first hypothesis) and clinical views in prescribing PrEP as a "best first step" across different high risk scenarios (second hypothesis).

### Survey measures

We used published data and clinical opinion to develop a 17-item survey instrument to assess practitioner practices and preferences of PrEP and other prevention options for groups vulnerable to HIV (S1 File) [18–21]. We surveyed prescriber characteristics, including age, sex, race/ethnicity, location of practice, qualification/licensure, length of practicing independently, and specialty area. Key content domains included: 1) experience prescribing PrEP, 2) the primary population for which the respondent has previously prescribed PrEP (if at all) (including MSM, PWID, and heterosexual individuals behaviorally vulnerable to HIV, including sero-different couples seeking conception), 3) how often in the course of routine care does the respondent typically ask his/her patients about partners' HIV serostatus, and 4) general level of comfort in counseling patients on safer conception options. We also developed clinical scenarios to examine practitioner preferences in prescribing PrEP as a "best first step" for sero-different couples trying to conceive, MSM, and PWID. The survey included case vignettes followed by specific clinical options that asked practitioners to identify the first clinically recommended step for groups at high risk for HIV (Fig 1).

### Analysis

We calculated descriptive statistics including means and standard deviations (SDs) for continuous variables and counts with proportions for categorical variables. We employed multivariable logistic regression analysis for binary and collapsed ordinal outcomes (experience prescribing PrEP and practitioner responses indicating PrEP as the "best first step" to vignette questions; for the latter, we dichotomized participant responses into "considered PrEP" or "did not consider PrEP"), using a backward elimination process, and included all covariates that altered the association between physician groups and outcome by more than 10%. Control variables included age, sex, ethnicity/race, CME course location, qualification/licensure, number of years practicing independently, and specialty area. We examined interaction between physician specialty and covariates such as age, sex, and preferences prescribing across risk categories. We excluded respondents with missing outcome variables for each study objective

*The following case vignettes are followed by specific clinical options, none of which can be considered either correct or incorrect. Please choose <u>one</u> option as the <u>best first step</u> in your opinion.*

1.  A 30-year-old male who is HIV-negative and has sex with men with an unremarkable PMH is sexually active with his HIV-positive husband. The (+) partner is on cART but having challenges reaching an undetectable HIV viral load. What do you recommend as a first step?

    ☐ Use condoms
    ☐ Recommend initiating PrEP for HIV-negative partner
    ☐ Use both condoms and initiate PrEP for HIV-negative partner
    ☐ Other (please specify):

2.  A 30-year old female who is HIV-negative with an unremarkable PMH is hoping to conceive with her HIV-positive partner. The (+) partner is on cART with an undetectable HIV viral load. What do you recommend as a first step?

    ☐ Do not conceive
    ☐ Recommend initiating PrEP for HIV-negative woman
    ☐ Refer couple for Assisted Reproductive Technology services
    ☐ Other (please specify):

3.  A 30-year old male who is HIV-negative with a history of injection drug use reports sharing needles with injection partners of unknown sero-status. What do you recommend as a first step?

    ☐ Refer to harm reduction program only, such as safe needle exchange
    ☐ Refer to substance abuse treatment and harm reduction program such as safe needle exchange if exists
    ☐ Recommend initiating PrEP
    ☐ Other (please specify):

**Fig 1. Vignettes identifying practitioner "best first step" for groups vulnerable to HIV.** This figure depicts the case vignettes on the survey instrument.

(experience prescribing PrEP by practitioner specialty and practitioner opinions on offering PrEP as a "best first step" in different risk scenarios), representing less than 1% and 3% of the total sample, respectively. We used SAS® 9.4 for statistical analyses. The raw anonymous data file and statistical output for the analyses can be found in the S2 and S3 Files).

## Results

### Demographic characteristics

Of the 954 participants across the five CME courses, 728 responded (response rate, 76%). Of those who responded, 652 were independent prescribers (MD/DOs, NPs, PAs). We excluded

pharmacists from the study because they did not independently prescribe or dispense PrEP for this population when the survey was distributed. Of our narrowed cohort of 652 independent prescribers, 519 (80%) responded to the survey. Of our final sample of 519, 362 (70%) were physicians, 45 (9%) PAs, and 112 (21%) NPs. Response rates for physicians, PAs, and NPs were 77%, 82%, and 90% respectively. Respondents' average age was 49 years (SD +/- 12.05) with the majority identifying as female (57%) and white (58%). Length of time in practice varied, though the majority having practiced independently for more than 10 years (59%). Our sample included 38% of prescribers specializing in infectious disease, 27% in family medicine, 26% in internal medicine, and 9% in other specialties (Table 1).

**Table 1. Demographic and prescribing characteristics.**

| CHARACTERISTIC | N/Mean | SD or % |
|---|---|---|
| CME Course Location | 519 | 100% |
| *New York City, NY* | 200 | 39% |
| *San Francisco, CA* | 77 | 15% |
| *Los Angeles, CA* | 71 | 14% |
| *District of Columbia, DC* | 63 | 12% |
| *Chicago, Illinois* | 108 | 21% |
| Age *(Missing: 21)* | 49 | +/- 12 |
| Identified sex *(Missing: 5)* | | |
| *Male* | 222 | 43% |
| *Female* | 292 | 57% |
| Ethnicity/Race *(Missing: 11)* | | |
| *Hispanic/Latino* | 42 | 8% |
| *White* | 293 | 58% |
| *Black or African American* | 52 | 10% |
| *Asian* | 94 | 18% |
| *Native Hawaiian or Pacific Islander* | 1 | 0.20% |
| *Other [a]* | 26 | 5% |
| Qualification/license | | |
| *MD/DO* | 362 | 70% |
| *PA* | 45 | 9% |
| *NP* | 112 | 21% |
| Time in practice *(Missing: 7)* | | |
| *In training* | 38 | 7% |
| *< 5 years* | 86 | 17% |
| *5–10 years* | 82 | 16% |
| *11–20 years* | 130 | 25% |
| *21+ years* | 176 | 34% |
| Specialty *(Missing: 3)* | | |
| *Infectious disease* | 197 | 38% |
| *Internal medicine* | 133 | 26% |
| *Family medicine* | 141 | 27% |
| *Other [b]* | 45 | 9% |
| Experience prescribing PrEP *(Missing: 4)* | 276 | 54% |
| *Infectious disease* | 103 | 37% |
| *Internal medicine* | 82 | 30% |
| *Family medicine* | 70 | 25% |
| *Other* | 21 | 8% |

*(Continued)*

**Table 1.** (Continued)

| CHARACTERISTIC | N/Mean | SD or % |
|---|---|---|
| **Prescribed population** *(Missing: 2; N = 274)* | | |
| *MSM* | 232 | 85% |
| *PWID* | 12 | 4% |
| *Heterosexual (not trying to conceive)* | 67 | 24% |
| *Anticipated conception exposure* | 44 | 16% |
| **Comfort in counseling people living with HIV on safer conception** *(Missing: 11)* | | |
| *Strongly agree/agree* | 333 | 65% |
| *Neutral* | 104 | 20% |
| *Strongly disagree/disagree* | 71 | 14% |
| **Case vignette #1: MSM, best first step** *(Missing: 17)* | | |
| *Use condoms* | 53 | 11% |
| *Initiate PrEP* | 27 | 5% |
| *Condoms and PrEP* | 410 | 82% |
| *Other* | 12 | 2% |
| **Case vignette #2: Conception, best first step [c]** *(Missing: 17)* | | |
| *Do not conceive* | 4 | 1% |
| *Initiate PrEP* | 285 | 56% |
| *Refer to assisted reproduction* | 182 | 36% |
| *PrEP or assisted reproduction* | 21 | 4% |
| *cART alone, undetectable viral load* | 6 | 1% |
| *Other* | 4 | 4% |
| **Case vignette #3: PWID, best first step** *(Missing: 17)* | | |
| *Refer to harm reduction program only* | 48 | 10% |
| *Refer to harm reduction and substance use treatment programs* | 275 | 55% |
| *Initiate PrEP* | 117 | 23% |
| *Treatment programs + PrEP* | 59 | 12% |
| *Other* | 3 | 0.5 |

[a] Other: Including mixed race (chose 2 or more categories).

[b] Other: 18 respondents specified pediatrics, 9 OBGYN, and 18 did not specify.

[c] When we designed the clinical vignettes, the CDC recommended either PrEP or other safer conception methods *irrespective of* partner virologic suppression for HIV sero-different couples seeking conception [22].

## PrEP prescribing patterns

Of the 519 respondents, 515 answered the question about experience prescribing PrEP; of these, 54% (276/515) had prescribed PrEP. In unadjusted logistic regression analyses, there were no significant proportional differences between practitioner specialty areas and the proportion who report ever prescribing PrEP. After adjusting for salient control variables, however, internal medicine practitioners were 1.6 times more likely than infectious disease practitioners to have prescribed PrEP, albeit with borderline significance (95% CI: 0.99–2.60, p = .0524) (main confounder was age). Each additional year of age reduced the odds of prescribing PrEP by 4% (aOR: 0.96, 95% CI: 0.93–0.98, p = 0.0014). Those still in training were 0.30 times as likely as those with 20 or more years of experience to have prescribed PrEP (95% CI: 0.11–0.84, p = 0.0214), and women were 0.57 times as likely as men to have prescribed PrEP (95% CI: 0.39–0.84, p = 0.0044) (Fig 2).

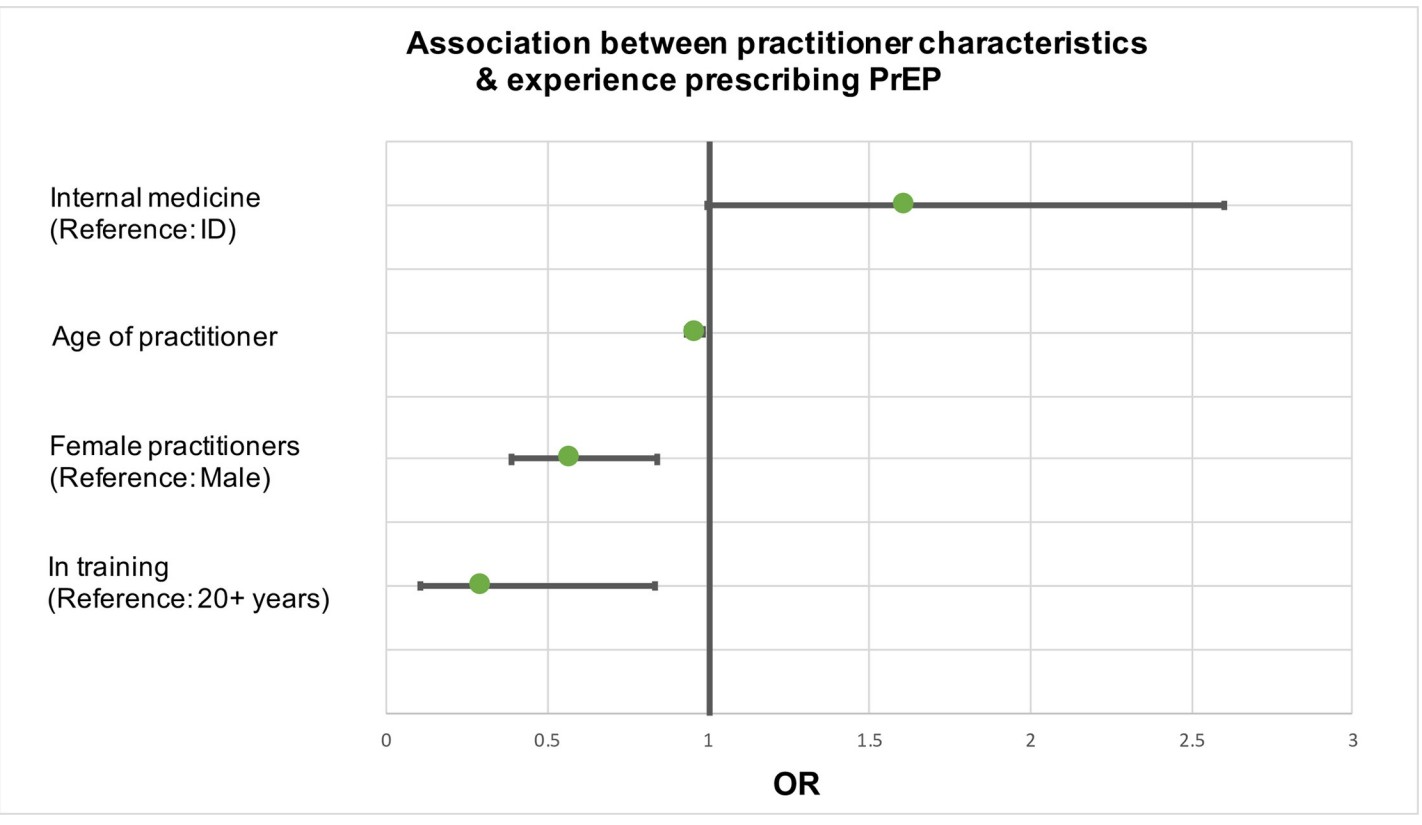

**Fig 2. Multivariable logistic regression analysis on the association of practitioner characteristics and experience prescribing PrEP.** This figure details the variables significantly impacting practitioner experience prescribing PrEP. ID = infectious disease.

## Willingness to prescribe PrEP as a "best first step" to different groups vulnerable to HIV

In our regression models when willingness to prescribe PrEP for the purpose of safer conception was the outcome variable, practitioners who viewed PrEP as the best first clinical step for MSM were less likely to have also considered PrEP as the first recommendation for safer conception purposes, albeit not significantly less likely (aOR: 0.750, 95% CI: 0.4–1.4, p = 0.356). On the other hand, practitioners who viewed PrEP as the first clinical step for PWID were twice as likely to have also considered PrEP as the best first step for safer conception, and vice-a-versa (95% CI: 1.4–3.2, p < .001) (Fig 3). Our tests of interactions were not statistically significant, and after adjusting for control variables, the model did not change.

In our regression models when willingness to prescribe PrEP for PWID was the outcome variable, practitioners who viewed PrEP as the first recommended step for MSM were nearly six times as likely to also consider PrEP as the best first step for PWID, and vice-a-versa (95% CI: 2.28–13.56, p = .0002). Our tests of interaction were not significant. When adjusting for important covariates in the model, our main results did not change. However, when specialty area was added, internal medicine practitioners were 1.94 times as likely (95% CI: 1.19–3.16, p = 0.0074) and family medicine practitioners two times as likely (95% CI: 1.26–3.33, p = 0.0037) as infectious disease specialists to view PrEP as the best first clinical step for PWID (Fig 3).

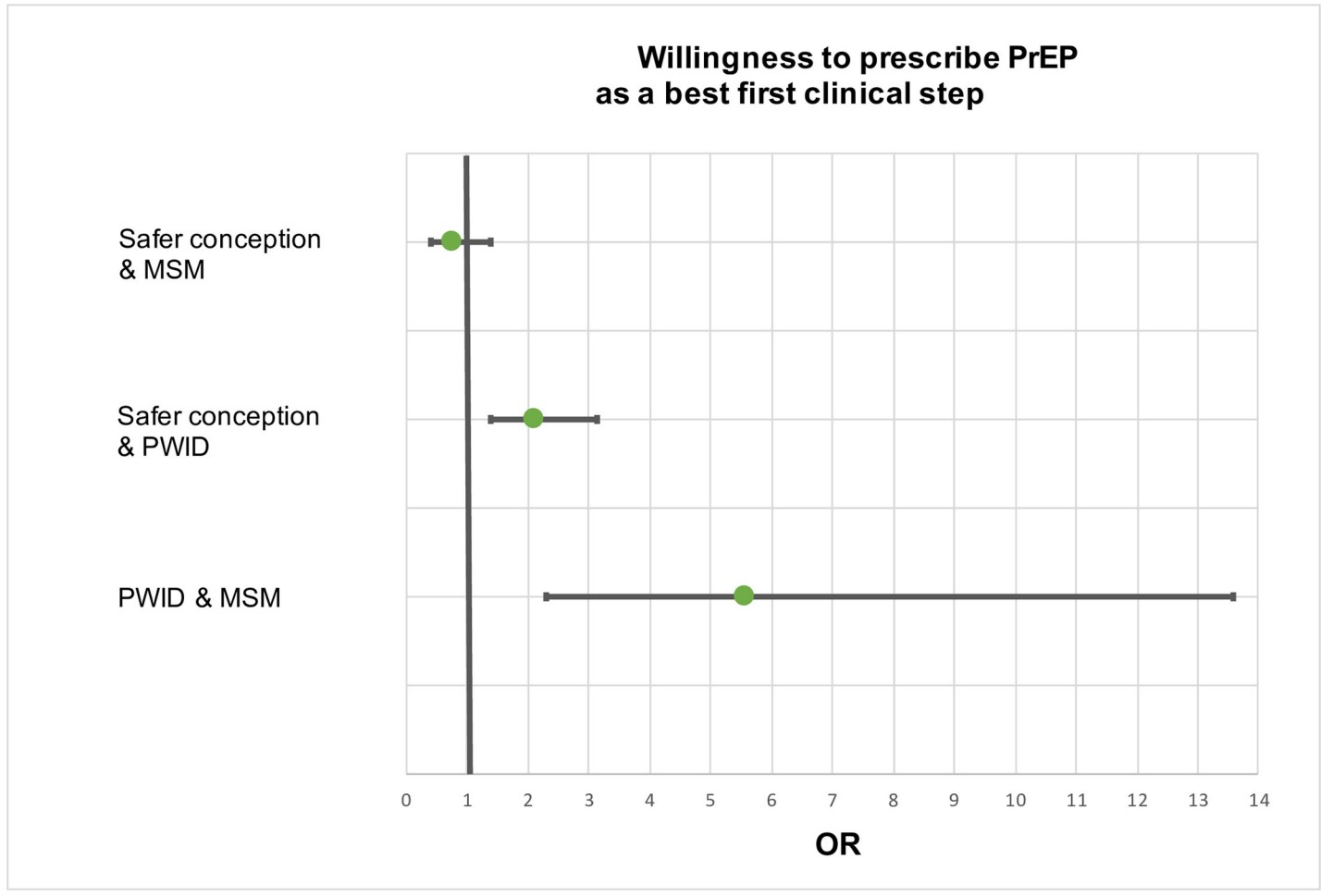

**Fig 3. Multivariable logistic regression analysis on willingness to prescribe PrEP as a best first clinical step across groups vulnerable to HIV.** This figure encompasses regression models comparing willingness to prescribe PrEP as a best first step in different risk scenarios–i.e. safer conception and MSM, safer conception and PWID, and PWID and MSM. MSM = Men who have sex with men; PWID = people who inject drugs.

## Discussion

The US Preventive Services Task Force has endorsed the value of PrEP for HIV prevention and recommended offering the medication to MSM, PWID, and heterosexually active men and women behaviorally vulnerable to HIV [2]. With less than 10% of individuals vulnerable to HIV taking the preventive medication in the US [2, 4, 5], our study characterizes a group of HIV-motivated clinicians that could play an important role in expanding PrEP availability in the US.

One important finding of this work is that among this highly motivated group of practitioners who were attending a continuing medical education course focused specifically on HIV, only 54% reported ever prescribing PrEP. Although this proportion is substantially higher than that reported in surveys of other practitioners, it may demonstrate that even among the most motivated clinicians, who are clearly focused on HIV care and seeking up-to-date information about HIV, only approximately half could be actively implementing current PrEP guidance. This finding could also reflect patients' own decision making around PrEP or practitioner lack of opportunity to offer PrEP. However, for PrEP to have a substantial impact on HIV incidence in the US, this proportion likely needs to increase. The low current rate of PrEP

prescribed by motivated clinicians suggests that they would be a high-value practitioner group to target for increased PrEP prescribing and tailored patient messaging.

Our finding that internal medicine practitioners were nearly twice as likely as infectious disease specialists to have experience prescribing PrEP supports the general consensus that primary care and family physicians play a crucial role in conducting HIV risk assessments and offering PrEP within their clinical purviews [6, 8, 9, 11]. Infectious disease specialists are also less inclined to encounter individuals without HIV compared to primary/family care practitioners. However, we know from recent literature that routine HIV risk assessments by clinicians is a barrier to implementing PrEP [11], which could further impact identifying and engaging with priority populations vulnerable to HIV. A study published in 2019 further reinforced barriers related to confusion or disagreement over the clinical purview for PrEP, with patients being "bounced back and forth between primary care and HIV clinics" [23]. Other barriers noted included knowledge gaps and practitioner attitudes, or stigma associated with PrEP prescribing.

Moreover, our findings suggest that practitioners might be more willing to prescribe PrEP as a best first clinical step for PWID than previously indicated [11, 14–16]. Practitioner specialty also mattered for the PWID case vignette, as internal and family medicine practitioners were twice as likely as infectious disease practitioners to view PrEP as an important first step for PWID. This result might suggest that infectious disease practitioners vary from other practitioners in prioritizing other preventive interventions before PrEP such as safe needle exchange programs or substance use treatment. Although our results indicating an inverse relationship between practitioner willingness to prescribe PrEP as the best first clinical step for MSM and for sero-different heterosexual couples seeking conception were statistically insignificant, this relationship should be further examined in future research [14, 24].

More work is needed to evaluate preventive interventions that engage priority groups vulnerable to HIV and potential barriers in identifying risk across populations and clinical specialties. Tailored messaging across different populations could pose an opportunity to better communicate with patients and reach individuals who could most benefit from PrEP [11, 17, 25].

## Limitations

Although the practitioners in our sample had a vested interest in HIV, this cohort likely represents the leaders driving PrEP availability across the country. Understanding their prescribing patterns and differentiation in prescribing across groups vulnerable to HIV could be important in expanding the use of PrEP. Although the missing responses to the outcome and control variables could have biased the results, no visible pattern existed in regard to the characteristics of omission, which comprised of less than 3% of the total sample. In addition, the vignettes and their response options were not parallel and therefore cannot be used to directly infer differences in PrEP prioritization by risk category. Finally, there may be important structural factors that could limit prescribing PrEP that are not examined in this study, including, but not limited to, patient financial barriers as a result of insurance restrictions and out of pocket costs.

## Conclusion

Heterosexual individuals and PWID make up more than a quarter of new HIV diagnoses each year in the US [26]. With a US policy goal to eradicate HIV by 2030, practitioners are central to ensuring the delivery of PrEP across care settings. Our findings indicate that even among a subset of HIV-focused practitioners, PrEP prescribing is not routine. Additionally, general

internal medicine practitioners were more likely to prescribe PrEP than HIV specialists, and practitioner willingness to prescribe PrEP varied across different risk scenarios. Our sample of motivated prescribers could be an optimal group to target to address any uncertainties in PrEP prescribing across risk groups and to effectively reach individuals that could most benefit from PrEP.

## Supporting information

**S1 File. Research survey conducted at IAS-USA Continuing Medical Education (CME) practitioner course.** This file comprises of the survey distributed at IAS-USA CME courses in five locations across the US (San Francisco, CA, Los Angeles, CA, New York, NY, Washington, DC, and Chicago, IL).
(DOCX)

**S2 File. Raw anonymous data file.** This csv file comprises of participant responses to the survey that is provided in the S1 File.
(CSV)

**S3 File. Statistical output.** This file encompasses the relevant statistical output for the multivariable logistic regression analyses presented in the Results section.
(DOCX)

## Author Contributions

**Conceptualization:** Ashley A. Leech, Benjamin P. Linas, Donna M. Jacobsen, Mari-Lynn Drainoni.

**Data curation:** Ashley A. Leech, Benjamin P. Linas, Donna M. Jacobsen, Isabel Morin, Mari-Lynn Drainoni.

**Formal analysis:** Ashley A. Leech, Cindy L. Christiansen, Benjamin P. Linas.

**Investigation:** Ashley A. Leech, Benjamin P. Linas, Donna M. Jacobsen, Mari-Lynn Drainoni.

**Methodology:** Ashley A. Leech, Cindy L. Christiansen, Benjamin P. Linas, Mari-Lynn Drainoni.

**Project administration:** Ashley A. Leech, Isabel Morin.

**Resources:** Ashley A. Leech, Donna M. Jacobsen.

**Software:** Ashley A. Leech.

**Supervision:** Cindy L. Christiansen, Benjamin P. Linas, Mari-Lynn Drainoni.

**Validation:** Ashley A. Leech, Cindy L. Christiansen, Benjamin P. Linas, Mari-Lynn Drainoni.

**Visualization:** Ashley A. Leech, Isabel Morin.

**Writing – original draft:** Ashley A. Leech.

**Writing – review & editing:** Ashley A. Leech, Cindy L. Christiansen, Benjamin P. Linas, Donna M. Jacobsen, Isabel Morin, Mari-Lynn Drainoni.

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
