## [Decision Letter · Decision Letter 0]

10 Mar 2020

PONE-D-19-35065

Healthcare practitioner experiences and willingness to prescribepre-exposure prophylaxis in the US

PLOS ONE

Dear Dr. Leech,

Thank you for submitting your manuscript to PLOS ONE. After careful consideration, we feel that it has merit but does not fully meet PLOS ONE’s publication criteria as it currently stands. Therefore, we invite you to submit a revised version of the manuscript that addresses the points raised during the review process.

Please respond to all of the reviewers' comments, with which I agree. Please focus in particular on thier comments on the study methods and the clarity of presentation of the methods, results, and conclusions. All of their comments will need to be addressed in order for the journal to consider this manuscript for publication. 

We would appreciate receiving your revised manuscript by Apr 24 2020 11:59PM. To enhance the reproducibility of your results, we recommend that if applicable you deposit your laboratory protocols in protocols.io, where a protocol can be assigned its own identifier (DOI) such that it can be cited independently in the future. For instructions see: http://journals.plos.org/plosone/s/submission-guidelines#loc-laboratory-protocols

We look forward to receiving your revised manuscript.

Kind regards,

Douglas S. Krakower, MD

Academic Editor

PLOS ONE

Journal Requirements:

2. Please provide additional details regarding participant consent. In the ethics statement in the Methods and online submission information, please ensure that you have specified (1) whether consent was informed and (2) what type you obtained (for instance, written or verbal). If your study included minors, state whether you obtained consent from parents or guardians. If the need for consent was waived by the ethics committee, please include this information.

3. In your Methods section, please provide additional information about the participant recruitment method and the demographic details of your participants. Please ensure you have provided sufficient details to replicate the analyses such as: a) the recruitment date range (month and year), b) a description of any inclusion/exclusion criteria that were applied to participant recruitment, c) a table of relevant demographic details, d) a statement as to whether your sample can be considered representative of a larger population, e) a description of how participants were recruited, and f) descriptions of where participants were recruited and where the research took place.

Reviewers' comments:

Reviewer's Responses to Questions

**Comments to the Author**

1. Is the manuscript technically sound, and do the data support the conclusions?

Reviewer #1: Partly

Reviewer #2: No

2. Has the statistical analysis been performed appropriately and rigorously? 

Reviewer #1: I Don't Know

Reviewer #2: No

3. Have the authors made all data underlying the findings in their manuscript fully available?

Reviewer #1: Yes

Reviewer #2: Yes

4. Is the manuscript presented in an intelligible fashion and written in standard English?

Reviewer #1: Yes

Reviewer #2: No

5. Review Comments to the Author

Reviewer #1: This manuscript presents self-reported PrEP experiences and attitudes from a large survey sample of providers attending HIV-focused CME courses across 5 major US cities. The manuscript has several strengths, including its large sample size and response rate, particularly for a provider survey. I also appreciated the authors’ initial framing of the introduction and discussion with respect to the new HIV policy and USPSTF recommendations. However, several aspects of the study dampened my overall enthusiasm for the work. I have several comments and suggestions for the authors’ consideration:

Major Comments/Suggestions

1. The vignette component of the study is problematic for several reasons:

a) The vignettes do not address the stated research aim corresponding to the vignettes (identifying risk groups perceived to be important)

To address the aim as stated, the survey would need to ask about perceived importance of the various groups for PrEP or willingness to prescribed PrEP to the groups. The vignette items in their current form only ask for the “best first step”; referring a person who injects drugs to a harm reduction program as a “best first step” before PrEP may be clinically appropriate for the patient’s immediate needs and does not tell us whether or not the provider perceives PrEP as important for this group (whether they may consider PrEP a second or concurrent step).

b) The vignettes also do not address the stated hypothesis corresponding to the vignettes (differences between practitioner willingness to prescribe PrEP across indications). To address the hypothesis about differences in prescription willingness across risk indications, the researchers would have needed to use parallel vignettes manipulating only risk indication. Instead, the vignettes vary in multiple ways, including nature of risk (serodiscordant male same-sex sexual activity with unknown drug use, serodiscordant heterosexual condomless sex with unknown drug use, or drug injection with unknown sexual risk); the HIV and viral load statuses of partners (positive/detectable, positive/undetectable, unknown); number of partners (one vs. multiple); the relationship status of partners (e.g., married vs. unspecified); and patients’ implied intentions related to future risk behavior (unclear vs. condomless sex vs. unclear). Furthermore, the response options are not parallel. For example, the MSM vignette offers condoms and PrEP+condoms as response options, but neither are options for the woman. Also, the MSM condition refers to PrEP as an option for an “HIV-negative partner” rather than the patient himself, which may be a point of confusion.

c) The analyses performed with the vignette data (logistic regressions examining the association between prioritization of PrEP in one vignette is associated with prioritization of PrEP in another) do not appear to address either the vignette-related research aim or the vignette related hypothesis.

Minor Comments/Suggestions

[Abstract]

2. The final analytic sample size should be specified in the abstract. Right now only the size of the participant pool is stated and can be easily mistaken for sample size.

3. While I understand the authors’ desire to highlight the non-significant trend related to differences in prescription experience by provider type (IM vs. ID) and appreciate that they included the p-value for transparency, I encourage them to delete the nonsignificant trend related to willingness to prescribe in the MSM scenario vs. conception scenario. Figure 2 suggests it was very far from being significant and any difference should not be interpreted as meaningful.

4. When discussing any vignette-based comparisons in the abstract, I think it’s important for the authors to appropriately contextualize (e.g., “Based on clinical vignettes describing different hypothetical prescribing scenarios…”)

[Introduction]

5. Given Comment #1 above, I highly recommend reframing the intro, research question, and hypotheses to match the analyses.

[Methods]

6. Given the centrality of the vignettes to the paper, I strongly recommend including them and the associated instructions in the main text rather than a supplemental appendix.

[Results]

7. I recommend reporting the aOR for the non-significant association between the MSM and conception vignette.

[Discussion]

8. “…only 54% reported ever prescribing PrEP. While this proportion is substantially higher than that reported in surveys of other providers, it demonstrates that even among the most motivated clinicians, who are clearly focused on HIV care and seeking up-to-date information about HIV infection, only approximately half are actively implementing current PrEP guidance.” This is not technically accurate, as providers who have not prescribed may have discussed PrEP and offered to prescribe to patients and patients declined. Also, there is the possibility that providers lacked opportunity to offer PrEP (e.g., because they only see patients living with HIV in their practice, not HIV-negative individuals).

9. The following text does not accurately reflect the findings and should be cut “Our results indicating… PWIDS than previously indicated.” Additionally, the text around targeting risk groups later in the paragraph seems irrelevant given that differences in risk groups were not actually assessed (and cannot be because of the differences in vignettes; see Comment #1.)

10. It’s a bit surprising that prioritization of PrEP in the MSM vignette was positively associated with prioritization of PrEP in the PWID vignette, and prioritization of PrEP in the PWID vignette was positively associated with prioritization of PrEP in the conception vignette, but there was no significant association between prioritization of PrEP in the MSM vignette and the conception vignette. What do the authors make of this?

11. “Finally, model-based estimates for age and length of practice… extrapolation.” It’s unclear to me what the authors are saying here, but it seems to suggest that they did not code their data correctly or treat them correctly in the analyses. I recommend rewording for clarity.

12. I’m not sure what the authors are referring to when suggesting financial restrictions as a barrier to prescribing. Do they mean patients not accepting prescriptions because of patients’ financial limitations? Or health centers not offering PrEP due to limited capacity/funding to support PrEP-related care?

13. An important limitation to add is acknowledgment that the vignettes and their response options were not parallel and therefore cannot be used to infer differences in PrEP prioritization by risk group (see Comment 1).

Reviewer #2: This is a very important topic, and given the limited data and information available about prescribing behaviors and practices among providers, this study has the potential to contribute important information. The authors conducted a relatively large survey among a sample of providers attending HIV CME conferences. Given that these providers "should" be among the most comfortable and experienced prescribing anti-retroviral medications, this is an important population to consider in understanding their views about prescribing PrEP.

The manuscript as written however would benefit from a more thorough editing, especially in regards to organization, being very explicit with phrases and definitions. There are many phrases used in the paper that are not clear to me what the authors mean (e.g., “HIV motivated” or “independent prescribers.” The introduction would benefit from being better framed to indicate explicitly why this population of providers were being studied/or the need/importance to study this set of providers. The methods need more details of what the variables were, the exact outcomes. Additionally, it would be helpful specify what the actual analytic sample was – although there are 519 participants, there seem to be missing data for most variables (although small), and then results being presented in the text are hard to follow due to shifting denominators. Additionally, for the regression models, I am assuming that all missing cases were not included in the models? Would be helpful clarify all of these issues, and use a consistent denominator. Also, the results are not clear – there are discrepancies between the figures and what is presented described in the text.

The authors should consider presenting all the output data (OR, CI) for all variables in the model (e.g., both the unadjusted and adjusted analysis).

Overall, this seems to be a commendable study - and could provide information that is sparse among providers. However, the methods and results need to be reworked/reorganized as it is difficult to follow and unclear why the results are presented in the way they are and if conclusions are supported by the data. I would encourage the authors to thoroughly editing the paper and check the analysis, and presenting more of the data as it could be helpful information to the field.

Abstract:

1. What is meant by “HIV motivated” – it would be helpful to define this explicitly

2. In the results, this sentence is not clear: “Practitioners who viewed PrEP as a first recommended clinical step for MSM were less likely to have also considered PrEP as the first recommendation for conception purposes, albeit not significantly less likely.”

3. What is meant by clinical step vs. first option?

4. The first sentence of the conclusion is not explicitly supported by th results presented – i.e., for PrEP to have a significant impact on incidence … as this study did not examine models of incidence following increased prescribing by this group of providers.

5. The 954 sample is misleading as it appears the analytic sample was also 954 (vs. 519 mentioned in the results section of the manuscript).

Financial Disclosures: Seems like the Grant # is incomplete (P30…).

INTRODUCTION:

It would be helpful to further explicitly delineate in the introduction why it was important to focus on “HIV motivated” prescribers, and how “HIV motivated” was determined and what this phrase means in this paper. (was there a scoring system, etc. used to determine motivation or assumption that they’re attending a CME course therefore should be automatically “motivated”?

METHODS:

1. The survey measures would benefit from being described in more detail, e.g., providing the actual text of the primary/secondary outcomes.

2. What does conditional scenarios mean? What are the scenarios conditioned upon?

3. Sentence starting at Line 117, “The fields included….” Could perhpas provide an example of a case vignette or options for recommended clinical steps.

4. It would be helpful to specific the exact outcomes being modeled/analyzed.

RESULTS

1. Who were not independent prescribers or what types of providers were excluded?

2. In Table 1, what is meant by “anticipated conception exposure”? may be helpful to define/mention these categories in the methods.

3. Was harm reduction and “substance use treatment program” defined for participants – as many times these could be the same, or that substance use treatment programs incorporate harm reduction – would be helpful to define these categories in the methods.

4. What are the “salient control variables”?

5. It is difficult to follow the willingness to prescribe paragraph. It also appears to be missing data for MSM?

6. It would be more clear and perhaps helpful to describe factors associated with willingness to prescribe and to then list the results in that way.

7. FIGURE 2. – there are two rows that show the same variable (willingness to prescribe PrEP as first best step for MSM” under outcome “willingness to prescribe for conception � which variable has the correct data – in this figure, it seems that willingness to prscribe for MSM IS associatd with willingness to prescribe for conception (rather than what is stated in the text).

Discussion - it may be that ID providers especially don’t see HIV-negative patients and therefore they’d never even have the opportunity to prescribe PrEP and thus efforts to “target” ID specialists may not necessarily be as fruitful given their lack of HIV-negative patients in their practices.

6. PLOS authors have the option to publish the peer review history of their article (what does this mean?). If published, this will include your full peer review and any attached files.

Reviewer #1: No

Reviewer #2: No

---

## [Author Response · Author response to Decision Letter 0]

16 Apr 2020

April 16, 2020 

To: Joerg Heber, PhD, Editor-in-Chief, PLOS One

& Douglas S. Krakower, MD, Academic Editor 

Dear Dr. Heber, Dr. Krakower, and the Editorial Board: 

Thank you for the opportunity to resubmit our revised manuscript titled, “Healthcare practitioner experiences and willingness to prescribe pre-exposure prophylaxis in the US” to PLOS One as a research article. We believe that our revisions based on the reviewers’ comments greatly improved the clarity and quality of our manuscript. Our detailed responses to each reviewer comment are enclosed. 

We hope that you will find our revised manuscript of high relevance and interest for your audience. Less than 10 percent of the more than one million people at risk for HIV are using pre-exposure prophylaxis (PrEP) in the US, and practitioners are critical to ensuring the delivery of PrEP across care settings. In our paper, we target a group of prescribers focused on providing HIV care and seeking up-to-date information about HIV and assessed their experiences prescribing PrEP, whether these experiences differed by clinical specialty, and how their clinical opinions on offering PrEP varied across risk scenarios. For PrEP to have a significant impact on HIV incidence in the US, the low rate of PrEP prescribing among HIV-focused clinicians likely needs to increase. This group of practitioners could be an optimal group to target to address any misgivings of PrEP prescribing across risk categories and to reach individuals that could most benefit from PrEP. 

We believe that our study will be of interest to your readership and an important topic that will impact policy and prioritization to reduce the spread of HIV in the US. Many thanks for your consideration. Please do not hesitate to contact me at ashley.leech@vanderbilt.edu if you have any questions. 

Kindest regards,

Ashley A. Leech, PhD, MS 

Assistant Professor

Department of Health Policy 

Vanderbilt University School of Medicine 

 ¬¬

JOURNAL / EDITORIAL CHANGES 

1. Please ensure that your manuscript meets PLOS ONE’s style requirements, including those for file naming. 

We have ensured that our manuscript meets PLOS ONE’s style requirements accordingly. 

2. Please provide additional details regarding participant consent. In the ethics statement in the Methods and online submission information, please ensure that you have specified (1) whether consent was informed and (2) what type you obtained (for instance, written or verbal). If your study included minors, state whether you obtained consent from parents or guardians. If the need for consent was waived by the ethics committee, please include this information.

We have added additional details to the Methods section regarding participant consent: 

Methods: [Study Design] "In collaboration with the International Antiviral Society-USA (IAS-USA), between March and May 2015, we circulated a voluntary paper survey to 954 participants at continuing medical education (CME) advanced-level HIV courses in five locations across the US: San Francisco, CA, Los Angeles, CA, New York, NY, Washington, DC, and Chicago, IL. We surveyed participants before the course session on PrEP. We did not provide incentives for survey completion. We included in our sample only practitioners who could independently prescribe, including physicians ((Doctors of Medicine [MDs]) or Doctors of Osteopathic Medicine [DOs]), nurse practitioners (NPs), and physician assistants (PAs). The Boston University Medical Center Institutional Review Board (IRB) determined this study to be HIPPA exempt in accordance with 45 CFR 46.101. As per IRB stipulations, we included a consent statement at the top of the survey form.

3. In your Methods section, please provide additional information about the participant

recruitment method and the demographic details of your participants. Please ensure you have provided sufficient details to replicate the analyses such as: a) the recruitment date range (month and year), b) a description of any inclusion/exclusion criteria that were applied to participant recruitment, c) a table of relevant demographic details, d) a statement as to whether your sample can be considered representative of a larger population, e) a description of how participants were recruited, and f) descriptions of where participants were recruited and where the research took place.

In our Methods section under [Study Design], we provide information on participant recruitment and demographic details of participants. We also specify the survey date range (month and year), description of inclusion/exclusion criteria, representativeness of the sample, where the recruitment/research took place, and demographic details which are outlined in Table 1. 

We have uploaded the minimal anonymized data set necessary to replicate our study findings as a supporting information file (S2 File), which we also added to the Manuscript text as shown below under “Supporting Information” after the References: 

S2 File: Raw anonymous data file. This csv file comprises of participant responses to the survey that is provided in the S1 File. 

REVIEWERS’ COMMENTS

 REVIEWER #1

R1, 1. The vignette component of the study is problematic for several reasons:

a) The vignettes do not address the stated research aim corresponding to the vignettes (identifying risk groups perceived to be important). To address the aim as stated, the survey would need to ask about perceived importance of the various groups for PrEP or willingness to prescribed PrEP to the groups. The vignette items in their current form only ask for the “best first step”; referring a person who injects drugs to a harm reduction program as a “best first step” before PrEP may be clinically appropriate for the patientʼs immediate needs and does not tell us whether or not the provider perceives PrEP as important for this group (whether they may consider PrEP a second or concurrent step). 

b) The vignettes also do not address the stated hypothesis corresponding to the vignettes (differences between practitioner willingness to prescribe PrEP across indications). To address the hypothesis about differences in prescription willingness across risk indications, the researchers would have needed to use parallel vignettes manipulating only risk indication. Instead, the vignettes vary in multiple ways, including nature of risk (serodiscordant male samesex sexual activity with unknown drug use, serodiscordant heterosexual condomless sex with unknown drug use, or drug injection with unknown sexual risk); the HIV and viral load statuses of partners (positive/detectable, positive/undetectable, unknown); number of partners (one vs. multiple); the relationship status of partners (e.g., married vs. unspecified); and patientsʼ implied intentions related to future risk behavior (unclear vs. condomless sex vs. unclear).

Furthermore, the response options are not parallel. For example, the MSM vignette offers condoms and PrEP+condoms as response options, but neither are options for the woman. Also, the MSM condition refers to PrEP as an option for an “HIV-negative partner” rather than the patient himself, which may be a point of confusion.

c) The analyses performed with the vignette data (logistic regressions examining the association between prioritization of PrEP in one vignette is associated with prioritization of PrEP in another) do not appear to address either the vignette-related research aim or the vignette related hypothesis.

Thank you very much for this thorough feedback. As noted in our detailed responses below, we believe we have addressed your concerns regarding the vignettes and how they are interpreted. We have also included limitations around this in the Discussion [Limitations] section, which we explicitly note that the vignettes and their response options were not parallel and therefore cannot be used to directly infer differences in PrEP prioritization by risk group. 

R1, 2. [Abstract] The final analytic sample size should be specified in the abstract. Right now only the size of the participant pool is stated and can be easily mistaken for sample size.

Thank you for this suggestion – we further specified our analytic sample in the Abstract by adding the following text in red below: 

Abstract: [Setting and methods] “Between March and May 2015, we circulated a paper survey to 954 participants ((652 of whom met our inclusion criteria of being independent prescribers and 519 of those (80%) responded to the survey)) at continuing medical education advanced-level HIV courses in five locations across the US on practitioner practices and preferences of PrEP.”

R1, 3. While I understand the authorsʼ desire to highlight the non-significant trend related to differences in prescription experience by provider type (IM vs. ID) and appreciate that they included the p-value for transparency, I encourage them to delete the nonsignificant trend related to willingness to prescribe in the MSM scenario vs. conception scenario. Figure 2 suggests it was very far from being significant and any difference should not be interpreted as meaningful.

We removed the statistically nonsignificant trend in the Abstract and Results sections related to willingness to prescribe in the MSM versus conception scenarios. We also rearranged the wording related to this result in the Discussion. The remaining text and changes are outlined below: 

Abstract [Results]: We removed the MSM/conception result and added the following text: “Practitioners who viewed PrEP as the first clinical step for PWIDs were twice as likely to have also considered PrEP as the first clinical option for conception, and vice-a-versa (95% CI: 1.4-3.2, p<.001). Practitioners considering PrEP as the first preventive option for MSM were nearly six times as likely to also consider PrEP as the first clinical step for PWIDs, and vice-a-versa (95% CI: 2.28-13.56, p=.0002).” 

Results: We removed the MSM/conception result in the first paragraph under the sub-heading, “Willingness to prescribe PrEP as a “best first step” to different risk categories” The paragraph now reads as follows: “In our unadjusted regression models when willingness to prescribe PrEP for the purpose of safer conception was the outcome variable, practitioners who viewed PrEP as the first clinical step for PWIDs were twice as likely to have also considered PrEP as the first clinical option for conception, and vice-a-versa (95% CI: 1.4-3.2, p<.001) (Figure 2). Our tests of interactions were not significant, and after adjusting for control variables, the model did not change.” 

Discussion: We changed the last two paragraphs to read as follows: “Moreover, our findings suggest that practitioners might be more willing to prescribe PrEP as a best first clinical step for PWID than previously indicated (11,14–16). Practitioner specialty also mattered, as internal and family medicine practitioners were twice as likely as infectious disease practitioners to view PrEP as an important first step approach for PWIDs. This result might suggest that infectious disease practitioners vary from other practitioners in prioritizing other preventive interventions before PrEP such as safe needle exchange programs or substance use treatment. Although our results indicating an inverse relationship between practitioner willingness to prescribe PrEP as the best first clinical step for MSM and for serodiscordant heterosexual couples seeking conception were statistically insignificant, it might indicate hesitation among prescribers in considering PrEP for conception than for the more familiar indication for MSM, which should be further examined in future research (14, 24). The positive relationship between conception and PWID aligned with our intuition that the group of practitioners who might be more willing to prioritize PrEP for the less common indications of conception and PWID could vary from those willing to prioritize PrEP for the more familiar indication of MSM, and therefore, have more harmonized responses. 

More work is needed to evaluate preventive interventions that target individuals in each high-risk category and potential barriers in identifying risk across populations and clinical specialties. Tailored messaging across different populations could pose an opportunity to better communicate with patients and reach individuals who could most benefit from PrEP (11,17,25).

R1, 4. When discussing any vignette-based comparisons in the abstract, I think itʼs important for the authors to appropriately contextualize (e.g., “Based on clinical vignettes describing different hypothetical prescribing scenarios…”)

As suggested, we added the following text to the Abstract [Results]: “Based on clinical vignettes describing different hypothetical prescribing scenarios, practitioners who viewed PrEP as the first clinical step for persons who inject drugs (PWID) were twice as likely to have also considered PrEP as the first clinical option for conception, and vice-a-versa (95% CI: 1.4-3.2, p<.001). Practitioners considering PrEP as their first preventive option for MSM were nearly six times as likely to also consider PrEP as the first clinical step for PWID, and vice-a-versa (95% CI: 2.28-13.56, p=.0002).” 

R1, 5. [Introduction] Given Comment #1 above, I highly recommend reframing the intro, research question, and hypotheses to match the analyses.

Abstract: [Background and objectives] “In this study, we target a group of prescribers focused on providing HIV care and seeking up-to-date information about HIV and assessed their experiences prescribing PrEP, whether these experiences differed by clinical specialty, and how their clinical opinions on offering PrEP as a “best first step” varied across risk scenarios.” 

Introduction: “In this study, we target a group of prescribers focused on providing HIV care and seeking up-to-date information about HIV and assessed their experiences prescribing PrEP, whether these experiences differed by clinical specialty, and how their clinical opinions on offering PrEP as a “best first step” varied across risk scenarios. We hypothesized that there would be differences in experience prescribing PrEP by practitioner specialty (infectious disease, internal medicine, family medicine) and between practitioner opinions on offering PrEP as a “best first step” across risk scenarios (MSM, at-risk heterosexuals seeking conception, and PWID).” 

 Methods: [Study design] “We used α = 0.05 for all statistical tests and set β at 0.20 (or a power of 0.80) when assessing sample size, which indicated adequate power at a total sample size of 450 to detect small effects (0.14-0.15) between prescriber categories (first hypothesis) and clinical views in prescribing PrEP as a “best first step” across different high risk scenarios (second hypothesis).”

 Methods: [Survey mesures] “We also developed clinical scenarios to examine practitioner preferences in prescribing PrEP as a “best first step” for conception, MSM, or PWID. The survey included case vignettes followed by specific clinical options that asked practitioners to identify the first clinically recommended step for patients across risk categories (Figure 1).” 

Results: We also changed one of the sub-headings in the Results section to read: “Willingness to prescribe PrEP as a “best first step” to different risk categories” 

R1, 6. [Methods] Given the centrality of the vignettes to the paper, I strongly recommend including them and the associated instructions in the main text rather than a supplemental appendix.

 Thank you for this suggestion. We included the vignettes as a Figure within the text as opposed to a supplemental appendix. We included the following in the Methods section and also re-labeled the other Figures (2 and 3) accordingly: 

Methods: [Survey measures] “Figure 1: Case vignettes identifying practitioner best first step for patients across risk categories. This figure depicts the case vignettes on the survey instrument.” 

R1, 7. [Results] I recommend reporting the aOR for the non-significant association between 

the MSM and conception vignette. 

As per your recommendation above (R1, 3), we removed the statistically nonsignificant trend in the Abstract and Results sections related to willingness to prescribe in the MSM versus conception scenarios. We touch on this statistically nonsignificant result in the Discussion section but do not include it within the Results as per the reviewer’s suggestion to not include a nonsignificant finding. 

R1, 8. [Discussion] “…only 54% reported ever prescribing PrEP. While this proportion is 

substantially higher than that reported in surveys of other providers, it demonstrates that even among the most motivated clinicians, who are clearly focused on HIV care and seeking up-to-date information about HIV infection, only approximately half are actively implementing current PrEP guidance.”

This is not technically accurate, as providers who have not prescribed may have discussed PrEP and offered to prescribe to patients and patients declined. Also, there is the possibility that providers lacked opportunity to offer PrEP (e.g., because they only see patients living with HIV in their practice, not HIV-negative individuals).

We recognize this limitation. We modified this language to indicate that this finding may demonstrate the active implementation of PrEP among HIV motivated clinicians, as “active implementation” could also include scenarios in which you describe above, i.e. the lack of opportunity to prescribe and/or patients declining. However, we also added further language to be inclusive of other explanations that you note above: 

Discussion: “Although this proportion is substantially higher than that reported in surveys of other providers, it may demonstrate that even among the most motivated clinicians, who are clearly focused on HIV care and seeking up-to-date information about HIV, only approximately half could be actively implementing current PrEP guidance.This finding could also reflect patients’ own decision making around PrEP or practitioner lack of opportunity to offer PrEP. However, for PrEP to have a significant impact on HIV incidence in the U.S., this proportion likely needs to increase. The low current rate of PrEP prescribing among motivated clinicians suggests that they would be a high-value practitioner group to target for increased PrEP prescribing and tailored patient messaging.”

R1, 9. The following text does not accurately reflect the findings and should be cut “Our results indicating… PWIDS than previously indicated.” Additionally, the text around 

targeting risk groups later in the paragraph seems irrelevant given that differences in risk groups were not actually assessed (and cannot be because of the differences in vignettes; see Comment #1.)

We added the following text in the Discussion to ensure both text phrases comply with the reviewer’s Comment #1: “Moreover, our findings suggest that practitioners might be more willing to prescribe PrEP as a best first clinical step for PWID than previously indicated (11,14–16).” “More work is needed to evaluate preventive interventions that target individuals in each high-risk category and potential barriers in identifying risk across populations and clinical specialties. Tailored messaging across different populations could pose an opportunity to better communicate with patients and reach individuals who could most benefit from PrEP (11,17,25).” – the latter text suggests that more work is needed to evaluate preventive interventions that target each high-risk category; it does not indicate that this is what we have done, which is in accordance with Comment #1. 

R1, 10. Itʼs a bit surprising that prioritization of PrEP in the MSM vignette was positively associated with prioritization of PrEP in the PWID vignette, and prioritization of PrEP in the PWID vignette was positively associated with prioritization of PrEP in the conception vignette, but there was no significant association between prioritization of PrEP in the MSM vignette and the conception vignette. What do the authors make of this?

 Despite the lack of a significant relationship between prioritization of PrEP in the MSM vignette to that of conception (and vice-a-versa), we did find an inverse relationship between the two, which might indicate providers feeling more hesitant in considering PrEP for conception than for the more familiar indication for MSM. However, it was more surprising (yet encouraging) to find that practitioners who viewed PrEP as the first recommended step for MSM were nearly six times as likely to also consider PrEP as the best first step for PWID. It was less surprising to find a relationship between prioritization of PrEP between conception and PWID since one could hypothesize that the group of providers who might be more willing to prioritize PrEP for these at-risk groups might vary from those willing to prioritize PrEP for a more familiar indication of MSM, and therefore, have more harmonized responses. 

 We have added the following text to the Discussion to reflect our above intuition: 

 Discussion: “Although our results indicating an inverse relationship between practitioner willingness to prescribe PrEP as the best first clinical step for MSM and for serodiscordant heterosexual couples seeking conception were statistically insignificant, it might indicate hesitation among prescribers in considering PrEP for conception than for the more familiar indication for MSM, which should be further examined in future research (14, 24). The positive relationship between conception and PWID aligned with our intuition that the group of practitioners who might be more willing to prioritize PrEP for the less common indications of conception and PWID could vary from those willing to prioritize PrEP for the more familiar indication of MSM, and therefore, lead to more harmonized responses.” 

R1, 11. “Finally, model-based estimates for age and length of practice… extrapolation.” Itʼs unclear to me what the authors are saying here, but it seems to suggest that they did not code their data correctly or treat them correctly in the analyses. I recommend rewording for clarity.

 Many thanks for pointing out the lack of clarity when discussing model-based estimates for age and length of practice. After revisiting, we agree that the language around this point is not very clear. We have amended it as follows: 

Discussion: [Limitations] “Finally, model-based estimates for age and length of practice should be interpreted with caution, for two reasons: First, length of time in practice was available in categorical format only; and second, there were practitioners up to the age of 41 years in the “in training” group and practitioners as young as 49 years in the “21+ years’ experience” group.”

R1, 12. Iʼm not sure what the authors are referring to when suggesting financial restrictions as a barrier to prescribing. Do they mean patients not accepting prescriptions because of patientsʼ financial limitations? Or health centers not offering PrEP due to limited capacity/funding to support PrEP-related care?

Thank you for pointing out this lack of clarification. We have added the following text accordingly: “Finally, there may be important structural factors that could limit prescribing that are not examined in this manuscript, including, but not limited to, patient financial barriers as a result of insurance restrictions and out of pocket costs.”

R1, 13. An important limitation to add is acknowledgment that the vignettes and their 

Response options were not parallel and therefore cannot be used to infer differences in PrEP prioritization by risk group (see Comment 1). 

We have added text within the limitation section of the Discsusion to address the reviewer’s concern regarding the vignettes: “In addition, the vignettes and their response options were not parallel and therefore cannot be used to directly infer differences in PrEP prioritization by risk category.”

 REVIEWER #2

R2, 1. This is a very important topic, and given the limited data and information available

about prescribing behaviors and practices among providers, this study has the potential to contribute important information. The authors conducted a relatively large survey among a sample of providers attending HIV CME conferences. Given that these providers "should" be among the most comfortable and experienced prescribing anti-retroviral medications, this is an important population to consider in understanding their views about prescribing PrEP.

The manuscript as written however would benefit from a more thorough editing, especially in regards to organization, being very explicit with phrases and definitions. There are many phrases used in the paper that are not clear to me what the authors mean (e.g., “HIV motivated” or “independent prescribers.” The introduction would benefit from being better framed to indicate explicitly why this population of providers were being studied/or the need/importance to study this set of providers. The methods need more details of what the variables were, the exact outcomes. Additionally, it would be helpful specify what the actual analytic sample was – although there are 519 participants, there seem to be missing data for most variables (although small), and then results being presented in the text are hard to follow due to shifting denominators. Additionally, for the regression models, I am assuming that all missing cases were not included in the models? Would be helpful clarify all of these issues, and use a consistent denominator. Also, the results are not clear – there are discrepancies between the figures and what is presented described in the text.

The authors should consider presenting all the output data (OR, CI) for all variables in the model (e.g., both the unadjusted and adjusted analysis).

Overall, this seems to be a commendable study - and could provide information that is sparse among providers. However, the methods and results need to be reworked / reorganized as it is difficult to follow and unclear why the results are presented in the way they are and if conclusions are supported by the data. I would encourage the authors to thoroughly editing the paper and check the analysis, and presenting more of the data as it could be helpful information to the field.

Thank you very much for this helpful feedback. As noted in our detailed responses below and to Reviewer 1, we believe we have addressed your concerns and recommendations as it relates to the clarity and structure of the piece. Thank you very much again for this thorough feedback. 

R2, 2. [Abstract] What is meant by “HIV motivated” – it would be helpful to define this explicitly

 We have changed the following text in both the Abstract and Introduction to clarify what we meant by “HIV motivated”: “In this study, we target a group of prescribers focused on providing HIV care and seeking up-to-date information about HIV and assessed their experiences prescribing PrEP”

R2, 3. [Abstract] In the results, this sentence is not clear: “Practitioners who viewed PrEP as a first recommended clinical step for MSM were less likely to have also considered PrEP as the first recommendation for conception purposes, albeit not significantly less likely.”

As per Reviewer #1’s suggestion (R1, 3), we removed this statistically nonsignificant finding from the Results section. 

R2, 4. [Abstract] What is meant by clinical step vs. first option?

 We removed “first option” from the Discussion and replaced with “first clinical step,” which is also synonomous with “best first step” in regards to practitioner prioritization of PrEP among other options of care across different at-risk categories. 

R2, 5. [Abstract] The first sentence of the conclusion is not explicitly supported by the 

results presented – i.e., for PrEP to have a significant impact on incidence … as this study did not examine models of incidence following increased prescribing by this group of providers. 

 Thank you for this comment. We are looking at the larger picture in regard to the impact that PrEP prescribing might have on HIV incidence in the US. However, we have replaced this sentence in the Conclusion in the Abstract with the following: “Our findings indicate that even among a subset of HIV-focused practitioners, PrEP prescribing is not routine.” We have kept this statement in the Discussion section of the manuscript since making broad inferences at the population level is important in understanding how the impacts of prescribing might have on overall HIV incidence in the US. 

R2, 6. [Abstract] The 954 sample is misleading as it appears the analytic sample was also 

954 (vs. 519 mentioned in the results section of the manuscript). 

 As per Reviewer 1’s comment (R1, 2), we further specified our analytic sample in the Abstract by adding the following text in red below: 

Abstract: [Setting and methods] “Between March and May 2015, we circulated a paper survey to 954 participants ((652 of whom met our inclusion criteria of being independent prescribers and 519 of those (80%) responded to the survey)) at continuing medical education advanced-level HIV courses in five locations across the US on practitioner practices and preferences of PrEP.”

R2, 7. Financial Disclosures: Seems like the Grant # is incomplete (P30…).

 Thank you for pointing this out – we will be sure to fix this in the revised submission. 

R2, 8. [Introduction] It would be helpful to further explicitly delineate in the introduction 

why it was important to focus on “HIV motivated” prescribers, and how “HIV motivated” was determined and what this phrase means in this paper (was there a scoring system, etc. used to determine motivation or assumption that theyʼre attending a CME course therefore should be automatically “motivated”? 

These are a group of prescribers that are attending an advanced-level HIV CME course who are focused on HIV care and clearly seeking up-to-date information on HIV. We have added the following text to the introduction to specify what we mean by “HIV-motivated” prescribers: “In this study, we target a group of prescribers focused on providing HIV care and seeking up-to-date information about HIV and assessed their experiences prescribing PrEP, whether these experiences differed by clinical specialty, and how their clinical opinions on offering PrEP as a “best first step” varied across risk scenarios.”

R2, 9. [Methods] The survey measures would benefit from being described in more detail, 

e.g., providing the actual text of the primary/secondary outcomes. 

 The questionnaire is added as a supplementary file within this section. In addition, as per Reviewer #1’s suggestion, we also included the vignettes as a Figure within the text as opposed to a supplemental appendix. We included the following in the Methods section and also re-labeled the other Figures (2 and 3) accordingly: 

Methods: [Survey measures] “Figure 1: Case vignettes identifying practitioner best first step for patients across risk categories. This figure depicts the case vignettes on the survey instrument.” 

R2, 10. [Methods] What does conditional scenarios mean? What are the scenarios conditioned upon?

We changed this statement to read: “We also developed clinical scenarios to examine practitioner preferences in prescribing PrEP as a “best first step” for conception, MSM, or PWID.”

R2, 11. [Methods] Sentence starting at Line 117, “The fields included….” Could perhpas 

provide an example of a case vignette or options for recommended clinical steps. 

As noted in R2,9 above and as per Reviewer #1’s suggestion, we also included the vignettes as a Figure within the text as opposed to a supplemental appendix. We included the following in the Methods section and also re-labeled the other Figures (2 and 3) accordingly: 

Methods: [Survey measures] “Figure 1: Case vignettes identifying practitioner best first step for patients across risk categories. This figure depicts the case vignettes on the survey instrument.” 

R2, 12. [Methods] It would be helpful to specific the exact outcomes being modeled / analyzed.

 Many thanks for this comment – we outline our outcomes and hypotheses under the Methods section [Study Design] in the last paragraph. However, we further specified the specific outcomes in the [Analysis] section as follows: “We employed multivariable logistic regression analysis for binary and collapsed ordinal outcomes (experience prescribing PrEP and practitioner responses indicating PrEP as the “best first step” to vignette questions),”

R2, 13. [Results] Who were not independent prescribers or what types of providers were excluded?

 “We added the following to the Results section [Demographic characteristics]: Of the 954 participants across the five CME courses, 728 responded (response rate, 76%). Of those who responded, 652 were independent prescribers (MD/DOs, NPs, PAs). We excluded pharmacists from the study because they did not independently prescribe or dispense PrEP for this population when the survey was distributed.”

R2, 14. [Results] In Table 1, what is meant by “anticipated conception exposure”? may be 

helpful to define/mention these categories in the methods. 

We have added the following clarifications in the Methods section [Survey Measures]: “We surveyed prescriber characteristics, including age, sex, race/ethnicity, location of practice, qualification/licensure, length of practicing independently, and specialty area. Key content domains included: 1) experience prescribing PrEP, 2) the primary population to which the respondent has previously prescribed PrEP (if at all) (including MSM, PWID, at-risk heterosexual individuals, and serodiscordant couples seeking conception), 3) how often in the course of routine care does the respondent typically ask his/her patients about partners’ HIV serostatus, and 4) general level of comfort in counseling patients on safer conception options.”

R2, 15. [Results] Was harm reduction and “substance use treatment program” defined for participants – as many times these could be the same, or that substance use treatment programs incorporate harm reduction – would be helpful to define these categories in the methods.

 We provide an example of what we mean by harm reduction in the vignette (i.e. “such as safe needle exchange”). As noted above, we included the vignettes as a Figure within the text as opposed to a supplemental appendix. 

R2, 16. [Results] What are the “salient control variables”?

Relevant control variables are specifically outlined in the Methods section [Analysis]: “Control variables included age, sex, ethnicity/race, CME course location, qualification/licensure, number of years practicing independently, and specialty area. We examined interaction between physician specialty and covariates such as age, sex, and preferences prescribing across risk categories.”

R2, 17. [Results] It is difficult to follow the willingness to prescribe paragraph. It also appears to be missing data for MSM?

As per Reviewer #1’s suggestion (R1, 3), we removed this statistically nonsignificant finding in the Results section, which we think adds clarity to this section. We additionally added the following to encompass the bidirectional relationship between between PWID and MSM: 

“In our unadjusted regression models when willingness to prescribe PrEP for PWID was the outcome variable, practitioners who viewed PrEP as the first recommended step for MSM were nearly six times as likely to also consider PrEP as the best first step for PWID, and vice-a-versa (95% CI: 2.28-13.56, p=.0002).”

R2, 18. [Results] It would be more clear and perhaps helpful to describe factors associated with willingness to prescribe and to then list the results in that way.

As noted in R2,9 above and as per Reviewer #1’s suggestion, we included the vignettes as a Figure within the text (in the Methods section) as opposed to a supplemental appendix. We hope that by adding the case vignettes identifying practitioner best first step for patients across risk categories will make the Results section easier to understand in regard to factors associated with willingness to prescribe. We also report on other factors in the Results section that are associated with willingness to prescribe PrEP such as specialty area. 

R2, 19. [Results] FIGURE 2. – there are two rows that show the same variable (willingness to prescribe PrEP as first best step for MSM” under outcome “willingness to prescribe for conception - which variable has the correct data – in this figure, it seems that willingness to prscribe for MSM IS associatd with willingness to prescribe for conception (rather than what is stated in the text). 

 ‘”Willingness to prescribe PrEP as ‘best first step’ for MSM’” is assessed both when “conception” is the outcome variable and when “PWID” is the outcome variable. Willingness to prescribe for MSM is NOT associated with willingness to prescribe for conception (as you can see it croses the 1 in the figure); however, as you can also see from the Figure, prescribing PrEP for MSM IS associated with willingness to prescribe for PWIDs. We have changed the Y-Axis labels to better clarify the odds ratios between “Conception & MSM,” “Conception & PWID,” and “PWID & MSM.” 

R2, 20. Discussion - it may be that ID providers especially donʼt see HIV-negative patients and therefore theyʼd never even have the opportunity to prescribe PrEP and thus efforts to “target” ID specialists may not necessarily be as fruitful given their lack of HIV-negative patients in their practices.

 Our sample of HIV motivated prescribers does not just consist of ID specialists, as we also surveyed internal, family, and other care providers. As we mention in the Discussion, we know from recent literature that routine HIV risk assessments by clinicians is a barrier to implementing PrEP (11), which could further impact identifying and targeting at-risk populations. A recent study by Skolnik and colleagues (2019) further reinforced barriers related to confusion or disagreement over clinical purview for PrEP, with patients being “bounced back and forth between primary care and HIV clinics” (22) – this purview issue has been a longstanding challenge that we discuss in the paper.

---

## [Decision Letter · Decision Letter 1]

9 Jun 2020

PONE-D-19-35065R1

Healthcare practitioner experiences and willingness to prescribe

pre-exposure prophylaxis in the US

PLOS ONE

Dear Dr. Leech,

Thank you for submitting your manuscript to PLOS ONE. After careful consideration, we feel that it has merit but does not fully meet PLOS ONE’s publication criteria as it currently stands. Therefore, we invite you to submit a revised version of the manuscript that addresses the points raised during the review process.

Acceptance will be conttingent upon revising the manuscript in accordance with the two reviewers' recommendations. 

We look forward to receiving your revised manuscript.

Kind regards,

Douglas S. Krakower, MD

Academic Editor

PLOS ONE

Reviewers' comments:

Reviewer's Responses to Questions

**Comments to the Author**

1. If the authors have adequately addressed your comments raised in a previous round of review and you feel that this manuscript is now acceptable for publication, you may indicate that here to bypass the “Comments to the Author” section, enter your conflict of interest statement in the “Confidential to Editor” section, and submit your "Accept" recommendation.

Reviewer #1: (No Response)

Reviewer #2: (No Response)

2. Is the manuscript technically sound, and do the data support the conclusions?

Reviewer #1: Partly

Reviewer #2: Partly

3. Has the statistical analysis been performed appropriately and rigorously? 

Reviewer #1: I Don't Know

Reviewer #2: Yes

4. Have the authors made all data underlying the findings in their manuscript fully available?

Reviewer #1: Yes

Reviewer #2: Yes

5. Is the manuscript presented in an intelligible fashion and written in standard English?

Reviewer #1: Yes

Reviewer #2: Yes

6. Review Comments to the Author

Reviewer #1: This manuscript presents a survey-based study of US providers’ PrEP attitudes and experiences. I reviewed an earlier version of the manuscript as Reviewer #1. Overall, I am pleased with the authors’ responsiveness to my and the other reviewer’s feedback and I believe the manuscript has been strengthened considerably. I have just a few remaining comments and suggestions, all of which are fairly minor:

1. P. 3/Lines 80-81: When first using the new language replacing “HIV-motivated” (i.e., “focused on providing HIV care and seeking up-to-date information about HIV…”) I suggest adding the following parenthetical clarification: “(as evidenced by their attendance of an advanced HIV continuing medical education course)”

2. There are 4 places where the wording still seems to imply that direct comparisons were made in providers’ judgments across risk scenarios. As noted in my original critique, this is not appropriate given that the vignettes and their response options were not parallel. Furthermore, this is not what was done in the analyses, which instead examined associations between perceptions of PrEP being a “best first step” in different risk scenarios. I highly recommend rewording the following:

(a) [Abstract] “…how their clinical opinions on offering PrEP as a “best first step” varied across risk scenarios”

(b) [Intro] P. 3/Lines 81-83: “assessed…. how their clinical opinions on offering PrEP as a “best first step” varied across risk scenarios.”

(c) [Intro] P. 3-4/Lines 83-87: “We hypothesized that there would be differences… between practitioner opinions on offering PrEP as a “best first step” across risk scenarios (MSM, at-risk heterosexuals seeking conception, and PWID).

(d) [Conclusion] P. 12/Lines 281-282: “…and practitioner willingness to prescribe PrEP varied across high-risk categories.” (also see my original critique about inferring “willingness” when not actually measuring it)

3. I believe one of my earlier comments (R1.3) may have been misinterpreted. I did not intend to suggest that non-significant findings be deleted from the Results section in the main text, but rather that a non-significant finding be omitted from the abstract because the wording suggested an association where there was none. In the main text, it is appropriate to specify the stats (OR, CI, and p) for the non-significant association between belief in PrEP as a 1st step for women and belief in PrEP as a 1st step for MSM.

4. On a related note, I agree with Reviewer 2’s suggestion that the authors include all statistical output for their unadjusted and adjusted logistic regressions. A table organizing this information might be a helpful addition.

5. P. 11/Lines 265-269: “Model-based estimates for age and length of time in practice should also be interpreted with caution, for two reasons: First, length of time in practice was available in categorical format only; and second, there were practitioners up to the age of 41 years in the “in training” group and practitioners as young as 49 years in the “21+ years’ experience” group”: I appreciate the authors’ attempt to clarify these limitations; however, I’m still having trouble understanding them. Re: the first, I assume time in practice was appropriately treated as a categorical variable in the statistical model? And so the author is saying that the categorical measurement approach that was used is less precise than if years in practice was measured as a continuous variable? It seems like this could be deleted as it doesn’t strike me as especially noteworthy. However, if the authors wish to retain it, I recommend clarifying. Re: the 2nd limitation, I don’t see why the variation in age within practice categories is a problem. Again, I would delete or clarify.

Reviewer #2: Overall, the paper is improved and authors have been responsive. There are still a few important issues to clarify/address as noted below, which should help strengthen the paper.

For the whole paper, I would suggest the authors use "people first language" whenever possible, consistent with recommendations from NIH, UNAIDS, and PLHIV advocacy organizations:

https://impaactnetwork.org/DocFiles/News/NIAID%20HIV%20Language%20Guide%20-%20March%202020.pdf

e.g., instead of high risk categories, can consider writing: groups at high risk

e.g., in DISUCSSION: consider changing “…all individuals in high-risk categories for HIV including MSM, PWID, and heterosexually active men and women with certain risk profiles (2).” To: “…MSM, PWID and heterosexually active men and women at high risk for HIV…”

e.g., at-risk populations could be changed to “people/groups at risk”

e.g., “targeting at-risk populations” could be changed to “ engaging populations/groups/individuals at risk”

e.g., “target individuals in each high-risk category”, could change to “ engage individuals at high risk”

e.g., “high-risk categories” � “groups at high risk”

METHODS:

While the survey instrument is included as a supp file, it would be helpful to the reader to include a sentence about how exacrtly the response categories were dichotomized for each of the vignettes, as it seems in some cases there are multiple options which include PrEP as part of the first step.

Line 139-140, it would be helpful to specify which outcomes the 1% and 3% refer to specifically.

In methods – the grouping of the following categories, seems odd: “conception, MSM, or PWID” as MSM and PWID refer to people and conception refers to an act. Consider replacing conception with people first language (e.g., individuals trying to conceive)

RESULTS:

Line 174-175, what was the aOR for age and the 95% CI?

It would be helpful to include at the least the output of the multivariable logistic regression as a supplementary file (e.g., what were the aOR and CI for the other categories of years in practice, other practitioners, etc.).

Figure 3: would be helpful to include reference groups for each category. As the last comment, it’d also be helpful to include the actual model output/results in a supplementary table if not the main paper.

DISCUSSION:

Lines 225-236

I think it would be helpful to explicitly mention that ID specialists are likely infrequently encountering individuals without HIV compared to IM/FM as an explanation for disparate prescribing. This issue may not be obvious to many readers and also there isn’t data about the extent to which ID specialists in this sample have patients without HIV or such patients referred to them. But, it is probable that there is a likely large difference between ID an FM/IM.

Lines 236 – 250.

While this paragraph is improved, based on the vignettes presented, it is not really possible to compare willingness to prescribe between categories. The Conception vignette, indicated a low risk situation (sero-different relationship AND undetectable viral load) compared to MSM (sero-different AND detectable viral load). The detectable viral load scenario makes these two vignettes very different and thus challenging to extrapolate results to make a head to head comparison. I think it would be an over-reach given the vignettes/results to interpret the data as “…it might indicate hesitation among prescribers in considering PrEP for conception than the more familiar indication for MSM…” Also, the model was statistically insignificant.

“This result might suggest that infectious disease practitioners vary from other practitioners in prioritizing other preventive interventions before PrEP such as safe needle exchange programs or substance use treatment.”

This sentence in the discussion is unclear overall, and particularly the phrase “harmonized responses.”

“…and PWID aligned with our intuition that the group of practitioners who might be more willing to prioritize PrEP for the less common indications of conception and PWID could vary from those willing to prioritize PrEP for the more familiar indication of MSM, and therefore, have more harmonized responses.”

7. PLOS authors have the option to publish the peer review history of their article (what does this mean?). If published, this will include your full peer review and any attached files.

Reviewer #1: No

Reviewer #2: No

---

## [Author Response · Author response to Decision Letter 1]

19 Jul 2020

July 19, 2020 

To: Joerg Heber, PhD, Editor-in-Chief, PLOS One

& Douglas S. Krakower, MD, Academic Editor 

Dear Dr. Heber, Dr. Krakower, and the Editorial Board: 

Thank you for the second opportunity to resubmit our revised manuscript titled, “Healthcare practitioner experiences and willingness to prescribe pre-exposure prophylaxis in the US” to PLOS One as a research article. We believe that our additional revisions based on the reviewers’ comments further improved the clarity and quality of our manuscript. Our detailed responses to each reviewer comment are enclosed. 

Many thanks again for this opportunity. Please do not hesitate to contact me at ashley.leech@vanderbilt.edu if you have any questions.

Kindest regards,

Ashley A. Leech, PhD, MS 

Assistant Professor

Department of Health Policy 

Vanderbilt University School of Medicine 

REVIEWERS’ COMMENTS 

(ROUND 2)

 REVIEWER #1

This manuscript presents a survey-based study of US providersʼ PrEP attitudes and experiences. I reviewed an earlier version of the manuscript as Reviewer #1. Overall, I am pleased with the authorsʼ responsiveness to my and the other reviewerʼs feedback and I believe the manuscript has been strengthened considerably. I have just a few remaining comments and suggestions, all of which are fairly minor:

R1, 1. P. 3/Lines 80-81: When first using the new language replacing “HIV-motivated” (i.e., “focused on providing HIV care and seeking up-to-date information about HIV…”) I suggest adding the following parenthetical clarification: “(as evidenced by their attendance of an advanced HIV continuing medical education course)”

 Thank you for this suggestion – we have added the following (now lines 100-101): 

 “In this study, we target a group of prescribers focused on providing HIV care and seeking up-to-date information about HIV (as evidenced by their attendance of an advanced HIV continuing medical education course) and…”

R1, 2. There are 4 places where the wording still seems to imply that direct comparisons were made in providersʼ judgments across risk scenarios. As noted in my original critique, this is not appropriate given that the vignettes and their response options were not parallel. Furthermore, this is not what was done in the analyses, which instead examined associations between perceptions of PrEP being a “best first step” in different risk scenarios. I highly recommend rewording the following:

(a) [Abstract] “…how their clinical opinions on offering PrEP as a “best first step” varied across risk scenarios”

(b) [Intro] P. 3/Lines 81-83: “assessed…. how their clinical opinions on offering PrEP as a “best first step” varied across risk scenarios.”

(c) [Intro] P. 3-4/Lines 83-87: “We hypothesized that there would be differences… between practitioner opinions on offering PrEP as a “best first step” across risk scenarios (MSM, at-risk heterosexuals seeking conception, and PWID).

(d) [Conclusion] P. 12/Lines 281-282: “…and practitioner willingness to prescribe PrEP varied across high-risk categories.” (also see my original critique about inferring “willingness” when not actually measuring it)

 We made the following changes as per your suggestions above: 

(a) [Abstract] - We assessed their experiences prescribing PrEP, whether these experiences differed by clinical specialty, and examined associations between willingness to prescribe PrEP as a “best first step” and different hypothetical prescribing scenarios.

(b) [Introduction] - We assessed their experiences prescribing PrEP, whether these experiences differed by clinical specialty, and examined associations between willingness to prescribe as a “best first step” and different hypothetical prescribing scenarios.

(c) [Introduction] – We hypothesized that there would be differences in experience prescribing PrEP by practitioner specialty (infectious disease, internal medicine, family medicine) and between practitioner opinions on offering PrEP as a “best first step” and different hypothetical risk scenarios ((MSM, sero-different heterosexuals seeking conception (comprised of one person with HIV and one without HIV), and PWID)). 

(d) [Conclusion] - Additionally, general internal medicine practitioners were more likely to prescribe PrEP than HIV specialists, and practitioner willingness to prescribe PrEP varied across different risk scenarios.

R1, 3. I believe one of my earlier comments (R1.3) may have been misinterpreted. I did not intend to suggest that non-significant findings be deleted from the Results section in the main text, but rather that a non-significant finding be omitted from the abstract because the wording suggested an association where there was none. In the main text, it is appropriate to specify the stats (OR, CI, and p) for the non-significant association between belief in PrEP as a 1st step for women and belief in PrEP as a 1st step for MSM.

 Many thanks for the further clarification. We have re-added this result back into the Results section, only, as follows: 

 “In our regression models when willingness to prescribe PrEP for the purpose of safer conception was the outcome variable, practitioners who viewed PrEP as the best first clinical step for MSM were less likely to have also considered PrEP as the first recommendation for conception purposes, albeit not significantly less likely (aOR: 0.750, 95% CI: 0.4-1.4, p=0.356). On the other hand, practitioners who viewed PrEP as the first clinical step for PWID were twice as likely to have also considered PrEP as the best first step for conception, and vice-a-versa (95% CI: 1.4-3.2, p<.001) (Figure 3).” 

R1, 4. On a related note, I agree with Reviewer 2ʼs suggestion that the authors include all statistical output for their unadjusted and adjusted logistic regressions. A table organizing this information might be a helpful addition.

 We have included all statistical output as a supporting/suplementary document (S3 File). 

“S3 File: Statistical output. This file encompasses the relevant statistical output for the multivariable logistic regression analyses presented in the Results section.” 

R1, 5. P. 11/Lines 265-269: “Model-based estimates for age and length of time in practice 

should also be interpreted with caution, for two reasons: First, length of time in practice was available in categorical format only; and second, there were practitioners up to the age of 41 years in the “in training” group and practitioners as young as 49 years in the “21+ yearsʼ experience” group”: I appreciate the authorsʼ attempt to clarify these limitations; however, Iʼm still having trouble understanding them. Re: the first, I assume time in practice was appropriately treated as a categorical variable in the statistical model? And so the author is saying that the categorical measurement approach that was used is less precise than if years in practice was measured as a continuous variable? It seems like this could be deleted as it doesnʼt strike me as especially noteworthy. However, if the authors wish to retain it, I recommend clarifying. Re: the 2nd limitation, I donʼt see why the variation in age within practice categories is a problem. Again, I would delete or clarify.

 Yes, your interpretation is correct. As per your suggestion, we deleted this “limitation” accordingly. 

REVIEWER #2

Overall, the paper is improved and authors have been responsive. There are still a few

important issues to clarify/address as noted below, which should help strengthen the paper.

R2, 1. For the whole paper, I would suggest the authors use "people first language" 

whenever possible, consistent with recommendations from NIH, UNAIDS, and PLHIV advocacy organizations: https://impaactnetwork.org/DocFiles/News/NIAID%20HIV%20Language%20Guide%20-%20March%202020.pdf

e.g., instead of high risk categories, can consider writing: groups at high risk

e.g., in DISUCSSION: consider changing “…all individuals in high-risk categories for HIV including MSM, PWID, and heterosexually active men and women with certain risk profiles (2).” To: “…MSM, PWID and heterosexually active men and women at high risk for HIV…”

e.g., at-risk populations could be changed to “people/groups at risk”

e.g., “targeting at-risk populations” could be changed to “ engaging populations/groups/individuals at risk”

e.g., “target individuals in each high-risk category”, could change to “ engage individuals at high risk”

e.g., “high-risk categories” � “groups at high risk”

 We really appreciate this suggestion – many thanks for the reference document and for the reminder to use people first language. We have reviewed the document and changed the language in the text accordingly, thank you! Some examples of these changes throughout the text include: 

 “At risk for HIV” � “vulnerable to HIV”

 “Target” � “engage”

 “With certain risk factors and/or engaging in risky behaviors” � “who are behaviorally vulnerable to HIV”

 “At risk for” � “vulnerable to HIV” 

 “At risk groups � “priority groups”

 “Serodiscordant” � “sero-different”

 “Patients across risk categories” � “groups at high risk for HIV”

 Etc. 

R2, 2. [Methods] – While the survey instrument is included as a supp file, it would be helpful to the reader to include a sentence about how exactly the response categories were dichotomized for each of the vignettes, as it seems in some cases there are multiple options which include PrEP as part of the first step.

 We agree. We have added the following text: Methods [Analysis] lines 173-174: “‘We employed multivariable logistic regression analysis for binary and collapsed ordinal outcomes (experience prescribing PrEP and practitioner responses indicating PrEP as the “best first step” to vignette questions; for the latter, we dichotomized participant responses into “considered PrEP” or “did not consider PrEP”),’”

R2, 3. [Methods] – Line 139-140, it would be helpful to specify which outcomes the 1% and 3% refer to specifically.

We have added the following text to further specify which outcomes the 1% and 3% refer to: Methods [Analysis] lines 180-181: We excluded respondents with missing outcome variables for each study objective (experience prescribing PrEP by practitioner specialty and practitioner opinions on offering PrEP as a “best first step” in different risk scenarios), representing less than 1% and 3% of the total sample, respectively.

R2, 4. [Methods] In methods – the grouping of the following categories, seems odd: “conception, MSM, or PWID” as MSM and PWID refer to people and conception refers to an act. Consider replacing conception with people first language (e.g., individuals trying to conceive)

 We agree – we have added the following: Methods [Survey Measures] Lines 149-150: “We also developed clinical scenarios to examine practitioner preferences in prescribing PrEP as a “best first step” for sero-different couples trying to conceive, MSM, or PWID.”

 We also defined “sero-different couples” in the introduction, Line 107-108: “We hypothesized that there would be differences in experience prescribing PrEP by practitioner specialty (infectious disease, internal medicine, family medicine) and between practitioner opinions on offering PrEP as a “best first step” in different risk scenarios ((MSM, sero-different heterosexuals seeking conception (comprised of one person with HIV and one without HIV), and PWID)). 

R2, 5. [Results] Line 174-175, what was the aOR for age and the 95% CI?

 We have added the following: [Results], now line 217: “Each additional year of age reduced the odds of prescribing PrEP by 4% (aOR: 0.96, 95% CI: 0.93-0.98, p=0.0014).”

R2, 6. It would be helpful to include at the least the output of the multivariable logistic 

regression as a supplementary file (e.g., what were the aOR and CI for the other categories of years in practice, other practitioners, etc.).

We included the output of the multivariable logistic regression analyses as a supplementary file (S3 File). 

R2, 7. Figure 3: would be helpful to include reference groups for each category. As the last comment, itʼd also be helpful to include the actual model output/results in a supplementary table if not the main paper.

 We revised Figure 3; however, we did not include “reference” categories like in Figure 2 since we did not have reference categories for this model. The interpretation of the model illustrated in Figure 3 is different than that illustrated in Figure 2 (as detailed in the text when describing such relationships). 

As noted in comment R2,6 above, we included the output of the multivariable logistic regression analyses as a supplementary file (S3 File). 

R2, 8. [Discussion] Lines 225-236, I think it would be helpful to explicitly mention that ID specialists are likely infrequently encountering individuals without HIV compared to IM/FM as an explanation for disparate prescribing. This issue may not be obvious to many readers and also there isnʼt data about the extent to which ID specialists in this sample have patients without HIV or such patients referred to them. But, it is probable that there is a likely large difference between ID an FM/IM.

 [Discussion], now lines 287-288 “Our finding that internal medicine practitioners were nearly twice as likely as infectious disease specialists to have experience prescribing PrEP supports the general consensus that primary care and family physicians play a crucial role in conducting HIV risk assessments and offering PrEP within their clinical purviews (6,8,9,11). Infectious disease specialists are also less inclined to encounter individuals without HIV compared to primary/family care practitioners...”

R2, 9. [Discussion] Lines 236 – 250, while this paragraph is improved, based on the vignettes presented, it is not really possible to compare willingness to prescribe between categories. The Conception vignette, indicated a low risk situation (sero-different relationship AND undetectable viral load) compared to MSM (serodifferent AND detectable viral load). The detectable viral load scenario makes these two vignettes very different and thus challenging to extrapolate results to make a head to head comparison. I think it would be an over-reach given the vignettes/results to interpret the data as “…it might indicate hesitation among prescribers in considering PrEP for conception than the more familiar indication for MSM…” Also, the model was statistically insignificant.

 We removed this sentence, as suggested. 

R2, 10. This sentence in the discussion is unclear overall, and particularly the phrase “harmonized responses.” “…and PWID aligned with our intuition that the group of practitioners who might be more willing to prioritize PrEP for the less common indications of conception and PWID could vary from those willing to prioritize PrEP for the more familiar indication of MSM, and therefore, have more harmonized responses.”

 We removed this sentence from the discussion.

---

## [Editor Report · Decision Letter 2]

17 Aug 2020

Healthcare practitioner experiences and willingness to prescribe

pre-exposure prophylaxis in the US

PONE-D-19-35065R2

Dear Dr. Leech,

We’re pleased to inform you that your manuscript has been judged scientifically suitable for publication and will be formally accepted for publication once it meets all outstanding technical requirements.

Kind regards,

Douglas S. Krakower, MD

Academic Editor

PLOS ONE
---

## [Editor Report · Acceptance letter]

25 Aug 2020

PONE-D-19-35065R2 

Healthcare practitioner experiences and willingness to prescribe
pre-exposure prophylaxis in the US 

Dear Dr. Leech:

I'm pleased to inform you that your manuscript has been deemed suitable for publication in PLOS ONE. Congratulations! Your manuscript is now with our production department. 

Kind regards, 

on behalf of

Dr. Douglas S. Krakower 

Academic Editor

PLOS ONE